# Hydration Status of Geriatric Patients Is Associated with Changes in Plasma Proteome, Especially in Proteins Involved in Coagulation

**DOI:** 10.3390/nu15173789

**Published:** 2023-08-30

**Authors:** Laura Hoen, Daniel Pfeffer, Johannes R. Schmidt, Johannes Kraft, Janosch Hildebrand, Stefan Kalkhof

**Affiliations:** 1Institute for Bioanalysis, Coburg University of Applied Sciences and Arts, Friedrich-Streib-Str. 2, 96450 Coburg, Germany; 2Division of Geriatrics, Klinikum Coburg GmbH, 96450 Coburg, Germany; 3Proteomics Unit, Department of Preclinical Development and Validation, Fraunhofer Institute for Cell Therapy and Immunology—IZI, Perlickstr. 1, 04103 Leipzig, Germany

**Keywords:** proteomics, geriatric nutrition, chronic dehydration, coagulation

## Abstract

Due to multifactorial reasons, such as decreased thirst and decreased total body water, elderly patients are vulnerable to dehydration. The study aims to investigate whether moderate dehydration or hyperhydration affects the blood proteome. Blood samples, medication, and bioelectrical impedance analysis (BIA) details were collected from 131 geriatric patients (77 women and 54 men aged 81.1 ± 7.2 years). Based on an evaluation by Bioelectrical Impedance Vector Analyses (BIVAs) of this cohort, for each hydration status (dehydrated, hyperhydrated, and control), five appropriate blood plasma samples for both males and females were analyzed by liquid chromatography–mass spectrometry (LC-MS). Overall, 262 proteins for female patients and 293 proteins for male patients could be quantified. A total of 38 proteins had significantly different abundance, showing that hydration status does indeed affect the plasma proteome. Protein enrichment analysis of the affected proteins revealed “Wound Healing” and “Keratinization” as the two main biological processes being dysregulated. Proteins involved in clot formation are especially affected by hydration status.

## 1. Introduction

Dehydration refers to a reduction in the total body water, which can be due to a water deficit (water loss dehydration) or salt and water deficit (salt loss dehydration) [1]. There are two different forms of dehydration considering the blood sodium level: hypernatraemic (high blood sodium levels) or hyponatraemic (low blood sodium levels) [2]. Mostly, dehydration is the outcome of disease and/or the effects of medication [1], but the dehydration risk increases with age [3]. This is because aging is accompanied by various physical changes, including reduced thirst or decreased total body water and inadequate water intake [3]. As the demographic change leads to an increasingly aging population (age > 65) [4], the correct diagnosis of dehydration is becoming more important and, at the same time, challenging since tools such as capillary refill time and abnormal skin turgor or respiratory pattern cannot be applied to the elderly because of the aging process [3,5]. In addition to these external examinations, dehydration can be diagnosed with laboratory testing of the blood; for example, the plasma urea–creatinine ratio is a good indicator of whether the kidney is healthy [5]. However, older adults often have a raised ratio related to renal failure, heart failure, or other age-associated diseases [5]. In addition, the sodium level can be measured, but this is a less sensitive indicator than osmolality. Serum or plasma osmolality is considered the gold standard for diagnosing dehydration, as it can indicate dehydration from as low as 1% water loss [5]. Serum osmolality is calculated from lab values for blood urea nitrogen, serum bicarbonate, creatinine, glucose, sodium, calcium, and potassium [1,5]. Nevertheless, osmolality has its limitations, as it was formerly used for acute dehydration [6,7,8]. However, older adults have a more chronic dehydration [9,10], which is associated with elevated markers of inflammation and coagulation [11].

The research aim of this study is as follows:I.To find out whether and, if so, to what extent moderate dehydration or hyperhydration affects the blood proteome;II.To find which proteins are differentially abundant and could serve as potential biomarkers;III.To investigate which functions these dysregulations are related to.

## 2. Material and Methods

### 2.1. Study Design

With the focus on an investigation of hydration-associated diseases and changes, a geriatric study (ethical approval of the ethics committee of the University of Applied Sciences Coburg to the project “Storage and analysis of blood samples from dehydrated and adequately hydrated patients” from 3 December 2019, Code: HC_Kalkhof_03122019) was conducted in 2020. This included 131 female patients and male patients (all non-smokers aged 65 to 98 years). The study center was the Department of Geriatrics of Prof. Dr. Kraft (Regiomed Klinikum Coburg, Coburg, Germany). Of these patients, body composition, including hydration status, was recorded with BIA. In addition, data on medication, diagnoses, and blood values were recorded. In addition, blood samples were taken for subsequent proteomic analyses.

The patients were not conditioned in the context of diets, sports, etc., and none of them were initially hospitalized because of their hydration status. The examinations and sampling took place within the framework of regular ambulatory examinations. If patients had several examinations in 2020, several samples were also included, which ultimately led to a total number of 272 collected samples (Appendix A). A total of 71 of the 131 patients were measured at least twice.

### 2.2. Bioelectrical Impedance Vector Analyses (BIVAs)

Bioelectrical Impedance Vector Analyses and classification of patients according to their hydration status were performed according to Hoen et al. [12].

The InBody S10 (InBody 770, Cerritos, CA, USA), a multifrequency (1, 5, 50, 250, 500, and 1000 kHz) four-point bioelectrical impedance device, was used for all BIA approaches as several patients were not able to walk. Analyses were performed a maximum of 1 h after blood sampling. BIVA values were calculated as described by Piccoli et al. [13]. The whole-body resistance was calculated as previously described with the values for the right arm, right leg, and trunk at 50 kHz measured with an electric current intensity of 80 µA [12,14]. The formulas provided by Piccoli were used for the BIVA [15]. As the reference population, the Coburg cohort containing 130 male and 115 female patients with a BMI between 25 and 28 kg/m^2^ and an age between 65 and 98 was applied as previously described [12].

### 2.3. Selection and Characteristics of Analysed Samples

From the total 272 blood plasma samples, 5 samples of each hydration status (hyperhydrated, dehydrated, and control) in both genders were selected for further analysis. Only the first sample of each patient was considered because prolonged hospitalization would be expected to normalize hydration status. Alongside the hydration status, the medical condition was crucial. Exclusion criteria were tumor diagnosis, missing limbs, and incomplete patient information (for example, missing medication). The controls were matched as well as possible by age to the other hydration states.

The average age and number of drugs prescribed are shown in Table 1. The main diagnosis of all the patients analyzed included mobility impairment (e.g., due to fractures); only one of the hyperhydrated men was admitted for cerebral infarction.

### 2.4. Preparation of Blood Samples for Proteomic Analyses

Samples were collected during standard blood collection in the Department of Geriatrics. They were kept cold (4 °C) until they were centrifuged at 2500× *g* for 30 min at 4 °C on the same day within 3 h after collection. Afterward, the supernatant (blood plasma) was transferred to 1.5 mL Tubes (Greiner, Kremsmünster, Austria) and frozen at −80 °C.

The top 14 abundant proteins of each sample were depleted according to the manufacturer’s protocol with the High Select^TM^ Top14 Abundant Protein Depletion Mini Spin Columns (Thermo Fisher Scientific, Waltham, MA, USA). Total protein quantitation was determined using the Pierce 660 nm Protein Assay (Thermo Scientific Fisher, Bremen, Germany). After depletion, a modified filter-aided sample preparation (FASP) was conducted [16]. An ultrafiltration using a Vivacon 500 filter unit (30 kDa MWCO) was performed. A corresponding volume of 30 µg of depleted plasma proteins of each sample was added to the filter, and the buffer was changed to 8 M urea in 0.5 M TCEP ([16] methods 3.2, steps 1–3). This was followed by a thiol-alkylation with 50 mM iodoacetamide (IAA) in the adjusted buffer ([16] methods 3.2, step 4), which was then incubated for 30 min without mixing in the dark. To remove the excess IAA, the filter was rinsed twice with 8 M urea in 0.5 M TCEP with pH 8.0. Then, the filter units were transferred to a new collection tube, and 10 µL of urea buffer and 80 µL of trypsin (with an enzyme-to-protein ratio of 1:50) in 50 mM TEAB buffer was added. The tubes were sealed with parafilm to prevent evaporation and incubated overnight at 37 °C, mixing at 600 rpm. After centrifugation, the collected peptide samples were stored at −80 °C.

Subsequently, the samples were labeled with the TMT11plex Label Reagents (Thermo Scientific Fisher, Germany) in accordance with the manufacturer’s protocol (section C peptide labeling) with the following changes: 27 µL of the appropriate 10-plex solution was mixed to the samples, and 5 µL of the quenching solution was used.

A Common Reference was generated from aliquots of all samples (except for 2, as too little material remained from these, which concerns one dehydrated female and one hyperhydrated male). Therefore, 10 µL were taken from each sample and pooled. A total of 200 µL of this reference was labeled with 80 µL of TMT^10^-131 and quenched with 6 µL quenching solution.

After labeling, the samples were analyzed by liquid chromatography–mass spectrometry (LC-MS) as previously described [17]. Briefly, 1 µg of labeled peptides were injected into an Easy-nLC 1200 coupled to a Q Exactive HF mass spectrometer (Thermo Fisher Scientific, Germany). The peptides were separated on a 20 cm analytical HPLC column packed in-house using ReproSil-Pur C18-AQ 1.9-μm silica beads (Dr. Maisch GmbH, Ammerbuch, Germany). Separation was achieved by applying a 120 min multistep gradient from 10% solvent A to 90% solvent B at a constant flow rate of 200 nL/min. As mobile phases, 0.1% formic acid in water (solvent A) and 80% ACN, supplemented with 0.1% formic acid in water (solvent B), were applied. The eluting peptides were ionized by electrospray ionization at 3.2 kV. Data were acquired in data-dependent mode and controlled by XCalibur software (version 2.9). The survey scans were acquired in a scan range of 300–1650 *m*/*z*, a mass resolution of 60,000, an AGC target of 3 × 10^6^, and a maximum injection time of 25 ms. The top 10 most abundant precursor ions were selected for isolation (window of 0.7 amu) and fragmentation by higher-energy collisional dissociation (normalized collision energy: 34). For MS2 scans in orbitrap mass analyzer, AGC target and maximum injection time were set to 1 × 10^5^ and 110 ms, respectively. A dynamic exclusion for MS2 scans was set to 30 s. The proteomics raw data were analyzed using MaxQuant software (version 1.6.2.3), and the integrated protein identification algorithm Andromeda was utilized. The experimental spectra were matched against reference proteome of H. sapiens. Trypsin was defined as a position-specific protease in fixed mode with a tolerance of up to two missed cleavages. Carbamidomethylation (cysteine) was set as fixed modification, whereas oxidation (methionine) and acetylation (protein N-terminus) were set as optional modifications. A mass tolerance of 20 ppm (first search and fragment ions) and 4.5 ppm (main search) was allowed. Two peptides, including at least one unique peptide, were required for identification, controlling the false discovery rate (FDR) to 0.05 for peptide spectrum matches, and protein identification. The identified proteins were filtered to exclude potential contaminants and decoy entries.

### 2.5. Data Analysis

The proteomics MS data were deposited in the ProteomeXchange Consortium via the PRIDE [18] partner repository with the dataset identifier PXD043728.

Samples with less than 200 protein quantity values greater than zero (valid values) and significant sample outliers according to the Grubbs test were not included in further analysis. Each criterion excluded one of the hyperhydrated samples (both female and male). A minimum of three remaining samples in each group (hyperhydrated, dehydrated, and control) was ensured.

A common reference scaling was performed first, followed by a median adjustment using R (version 3.5.1). The means of each group (hyperhydrated vs. control, dehydrated vs. control, and dehydrated vs. hyperhydrated) were then tested for significance (*p* < 0.05) using a two-tailed *t*-test.

The significant differentially abundant proteins (DAPs) were then further analyzed using STRING to identify affected biological processes. Enrichment analysis was used to identify the biological processes involving most of the proteins in the apparent clusters.

## 3. Results 

### 3.1. Protein Abundance Is Linked with Hydration Status

A total of 272 blood samples were collected from 131 patients. Further, bioelectrical impedance analyses were performed for every patient within one hour after blood collection. The average age of the patients was 81.1 ± 7.2 years. A total of 54 patients were male (41.2%), whereas 77 were female (58.8%). Based on BIVAs considering our previously published reference cohort [12], the samples were assigned to a hydration status (hyperhydrated patients in the lower left and dehydrated patients in the upper right—Figure 1). The groups range from highly dehydrated (14 samples, 8♀ and 6♂), moderately dehydrated (22 samples, 12♀ and 10♂), to slightly dehydrated (9 samples, 4♀ and 5♂). A total of 77 samples (45♀ and 32♂) are normally hydrated. A total of 33 samples were highly hyperhydrated (18♀ and 15♂), 22 samples (15♀ and 7♂) were moderately hyperhydrated, and 16 samples (9♀ and 7♂) were slightly hyperhydrated. Because hydration status changes during hospitalization, only the first examination of each patient was considered for the proteomic approach because it is expected that hydration status should return to normal during hospitalization. Due to this exclusion criteria, only five out of seven male dehydrated patients were suitable for further investigations. Therefore, each group (dehydrated, hyperhydrated, and control) in each gender was delimited to five samples to keep the groups the same size for the statistics. Nine female patients were dehydrated on their first examination. None of these patients were initially hospitalized because of dehydration, which underlines the aspect of chronic dehydration in patients. A total of 9 male and 12 female patients were hyperhydrated, and 26 female and 17 male patients were normally hydrated on the first day of examination.

After common reference scaling and median adjustment, 262 proteins in the female patients and 293 proteins in the male patients could be quantified, of which 20 are significantly differentially abundant (*p* < 0.05) in female patients, and 23 are significantly differentially abundant (*p* < 0.05) in male patients. Five of these differentially abundant proteins (DAPs) are significant in both genders. Overall, 38 proteins were significantly altered in their abundance according to their *p*-value (Table 2). For each of these DAPs, the effect size, confidence intervals, and standard deviation were calculated (Appendix A). As the significance of each DAP between the different hydration statuses (dehydrated vs. control, hyperhydrated vs. control, and dehydrated vs. hyperhydrated) was tested, the amount of significant DAP differs in these groups (Table 2). Some DAPs are significant for more than one of the tested contrasts.

### 3.2. DAPs Are Involved in Wound Repair and Blood Coagulation Are Associated with Hydration

Protein–protein interaction analyses of DAPs using STRING [19] resulted in a highly significant clustering (PPI enrichment value of *p* < 1.0 × 10^−16^) consisting of 34 nodes and 126 edges (Figure 2). Overall, 63 biological processes from Gene Ontology with a false discovery rate (FDR) < 10^−3^, at least five involved genes, and *p* < 0.005 were covered by the network of DAPs. In total, 16 main processes could be identified (subtermini, processes included by these main processes, were excluded) (Table 3). Two subtermini of these biological processes (response to wounding and keratinization) cover most of the proteins in the two clusters, which were built according to the STRING analysis (Figure 2). Beyond the wound healing process and keratinization, blood coagulation, negative regulation of blood coagulation, fibrin clot formation, regulation of body fluid levels, or blood coagulation intrinsic pathway, which would cover some more of the DAPs, could be found significant (*p* < 0.005) in the network (not shown because of redundancy in the main proteins involved). Prothrombin and Antithrombin-III were significantly lower in dehydrated men vs. hyperhydrated men and dehydrated women vs. control. The fibrinogen gamma chain was significantly higher in dehydrated men vs. hyperhydrated men. In addition, coagulation factors V and IX were found to be significantly lower in dehydrated women vs. control (Figure 3). Furthermore, keratins, which are the key structural material for the outer layer of the skin, were found to be significantly higher in hyperhydrated men: Keratin Type I cytoskeletal 14 and Keratin Type II cytoskeletal 5. In addition, Hornerin (HRNR, a component of the epidermal cornified cell envelope) was significantly more abundant in hyperhydrated men vs. control.

As coagulation could be influenced by anticoagulants, we had a closer look at them. In total, 47 patients were receiving anticoagulant drugs (Apixaban, Edoxaban, Rivaroxaban, Heparin, and Phenprocoumon), corresponding to 112 of 272 samples taken, of which 8 were classified as dehydrated, 38 were classified as hyperhydrated, and 21 were classified as control. Out of the five proteomic samples chosen, there was one dehydrated female, two dehydrated men, two female controls, three male controls, no hyperhydrated female, and two hyperhydrated men (one of whom was not considered in the analyses because of the criteria (2.4)) who were taking anticoagulant drugs. Acetylsalicylic acid (ASA), in a dose of a maximum of 100 mg per day, can be used as a platelet aggregation inhibitor as well [20]. A total of 74 samples were receiving ASA as a platelet aggregation inhibitor, from which 1 was dehydrated, 11 were hyperhydrated, and 26 were controls.

**Figure 3 nutrients-15-03789-f003:**
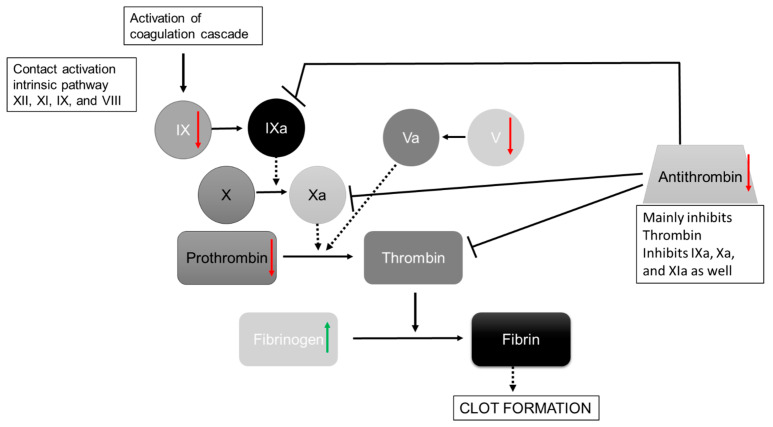
Simplified Illustration of the coagulation cascade after Bauer et al. [21]. Significant regulated proteins in the dehydration group summarized for men and women are shown with green arrows (=upregulated in dehydration state) and red arrows (=downregulated in dehydration state). Dashed arrows show the influencing proteins to the activation, and continuous arrows show activation pathways; inhibition is marked with a dead-end line, and coagulation factors are abbreviated with Roman numerals.

### 3.3. Alzheimer Prediction Marker

Pregnancy zone protein (PZP), as a suggested marker in Alzheimer’s disease [22,23], was also found to be differentially abundant. It was enriched in dehydrated men compared to hyperhydrated men (*p* = 0.006) and in hyperhydrated women compared to dehydrated women (*p* = 0.039). If the same criteria for dementia are applied as before [12], i.e., receiving memantine, donepezil, Ginkgo biloba extract, rivastigmine or striking Mini-Mental State Examination (MMSE) (Score ≤ 23), and/or Clock-Drawing Test (Score ≥ 3), then two of the five dehydrated men had dementia, as did two of the three hyperhydrated men. No hyperhydrated or dehydrated women in the samples analyzed had dementia.

## 4. Discussion

### 4.1. Plasma Proteome Is Affected by Moderate Dehydration and Hyperhydration

This study of 30 patients revealed 38 differential abundant proteins according to the hydration status (dehydrated, hyperhydrated, and normally hydrated) based on proteomic results. Furthermore, to the best of our knowledge, this is the first plasma proteome dataset in moderately dehydrated and hyperhydrated geriatric patients. The selection strategy of hydration status by BIVAs presented a representative subset for proteomics, although geriatric patients are affected by different diseases (average drug prescription of 10.5). This variance between the groups is also reflected by high standard deviations of the protein quantities within the groups. Beyond this, it should be considered that previous studies showed a lack of agreement between different BIA devices [24,25]. Although the reference population used is measured with a standing BIA (InBody 770) [12] and the patients in this study are measured with a supine BIA (InBody S10), both analyzers are multifrequency devices, whereas the reported differences between bioimpedance analyzers are mainly due to different measurement methods such as multifrequency, single frequency, and bioimpedance spectroscopy rather than standing or supine. Nevertheless, the different placement of the electrodes can have an impact, which should be kept in mind and has to be investigated further. However, even considering the patient and device-related variance, the proteome analysis of the blood plasma samples identified 38 DAPs, which indicates that the protein expression is not random and the hydration status has an influence on the abundance of blood plasma proteins. In turn, the DAPs identified interact in a highly interconnected network with significant pathways of hydration status. As several of the DAPs for dehydration, as well as hyperhydration, are affected in the same direction, e.g., F2 is for the females lower in both groups than in the control, the DAPs act as markers for general hydration status changes rather than the consequence of specific hydration direction.

### 4.2. Wound Repair/Coagulation Is Affected by Hydration Status

As already known, hydration status is an important factor in wound healing [26,27,28]. This study revealed a coherence between the hydration status and the wound healing process, especially the coagulation pathway, which is one of the main processes in wound healing [29]. Two coagulation factors (V, IX), as well as prothrombin, were found to be downregulated in dehydrated patients, which leads to an upregulation of Fibrinogen as it cannot be converted into fibrin (Figure 3). The increase in Fibrinogen in dehydrated humans has also been shown before [11]. Certainly, Antithrombin-III (ATIII) is also less abundant in dehydrated patients. Nevertheless, antithrombin deficiency can occur due to other conditions such as congenital antithrombin deficiency, liver damage, sepsis, renal diseases, consumption coagulopathy, ingestion of ovulation inhibitors, heparin therapy, and vitamin K deficiency [30]. Likewise, anticoagulation drugs inhibit the measurement of ATIII [30]. The patients analyzed did not have sepsis, a known vitamin K deficiency, or a known congenital antithrombin deficiency and did not take ovulation inhibitors. Only one of the dehydrated female patients has a liver disease, which should not have had a significant impact. The patient data showed that there was no difference in renal diseases (according to the criteria by Hoen et al. [12]) in the female patients (three out of five controls and dehydrated patients, as well as two out of the three analyzed hyperhydrated female samples). The renal diseases differ in four out of five dehydrated male patients, but only one of three hyperhydrated males was affected, and two of five controls had renal disease. Furthermore, some of the patients analyzed were given oral anticoagulant drugs, which also included heparin (3.2). As is already known, heparin activates antithrombin [31]. Heparin was taken by one hyperhydrated male, one control male, two control females, and one dehydrated female patient. Nonetheless, the difference of one patient should not influence the significance. Overall, more hyperhydrated samples (38 in total) than dehydrated samples (8 in total) were receiving anticoagulant drugs, which is in line with our previous study, where Apixaban, one of the anticoagulant drugs, was taken significantly more often in hyperhydrated patients [12].

In total, four of five dehydrated females, two of three hyperhydrated females, and three of five control females had at least one state that could influence ATIII levels, whereas four of five dehydrated men, two of three hyperhydrated men, and three of five control men had the same. In conclusion, the interaction of different influences of ATIII may play a role in the measurement of ATIII. Moreover, an increased consumption of ATIII could appear, which also leads to ATIII deficiency [32]. In addition, all of the patients analyzed were operated on prior to the blood sampling. As Büller et al. [33] showed, surgery induces a decrease in ATIII plasma levels to the lowest point at around the third postoperative day; this can also affect the measurements. Although all groups should be affected by this, dehydrated patients have lower levels than the other groups. In addition, Ruttmann et al. [34] reported that hemodilution results in decreased activity of ATIII in vitro. Taken together, multiple factors may interfere with the antithrombin abundance in patient’s plasma, especially in elderly cohorts suffering from comorbidities and increased drug administration. However, different studies have shown an effect of hydration status on the formation of venous thromboembolism [35,36] and deep vein thrombosis (DVT) [34,37,38], which underlines the relation of hydration status with the coagulation cascade.

### 4.3. Dehydration and Alzheimer

Ijsselstijn et al. [23] previously pointed out that serum levels of PZP are elevated in persons who were developing Alzheimer’s disease (AD) in comparison to persons who remained dementia-free. Dehydration decreases cognitive performance [39], which can lead to a further deterioration in the cognitive state. In addition, patients with AD have a major risk for current dehydration [40]. Dehydrated men showed increased levels of PZP, although the proportion of patients already diagnosed with dementia is higher in hyperhydrated men (66%) than in dehydrated men (40%). This is in line with our previous study, where dehydration is linked with Resveratrol, a medication used to improve mild cognitive impairment [12], and shows the unfavorable aspect of dehydration and cognitive decline. This was also observed by Allen et al., where a 48% decrease in dementia was predicted if the hydration was kept within normal levels [11]. Apart from that, PZP levels were increased in hyperhydrated women. The mutual deterioration is not given in this case, and it shows that PZP levels are not specifically linked with one hydration state.

### 4.4. Conclusions

In our study, we were able to show that the plasma proteome is indeed altered in patients who are found to be moderately dehydrated as well as hyperhydrated. In particular, the abundance of proteins functionally associated with wound healing and coagulation was found to be altered. In fact, dehydration leads to decreased coagulation, and hyperhydration is associated with increased coagulation, revealing proteins involved in the coagulation pathway as potential biomarkers. In addition, proteins such as prothrombin were differentially abundant in a gender-independent manner. Dehydration may also exacerbate cognitive decline, as the Alzheimer’s predictive marker (PZP) was found in dehydrated men. As a result, the study suggests that closer monitoring of hydration status is needed to prevent adverse aspects, which is in line with the findings of Allen et al. [11]. In this regard, plasma protein analysis may serve as a more sensitive and effect-related parameter compared to a general hydration analysis such as BIA.

## Figures and Tables

**Figure 1 nutrients-15-03789-f001:**
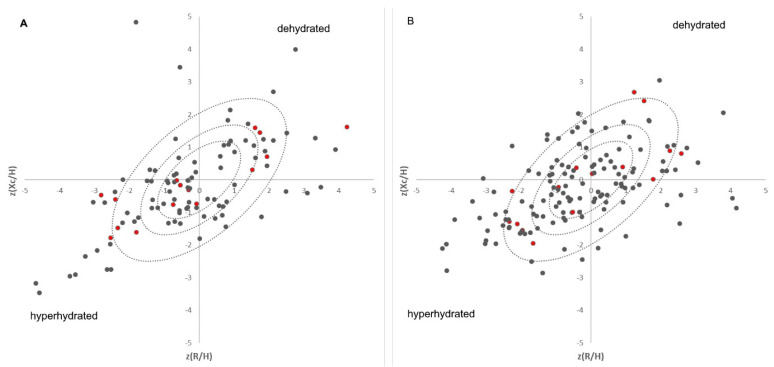
Distribution of the patients on each day of the sample collection in Bioelectrical Impedance Vector Analyses (male (**A**), female (**B**)) plotted against the tolerance ellipses of Coburg’s cohort [12]; Proteomic samples are highlighted in red, where the patients in the inner circle are the controls, the upper right ones are the dehydrated, and the lower left ones are the hyperhydrated patient samples.

**Figure 2 nutrients-15-03789-f002:**
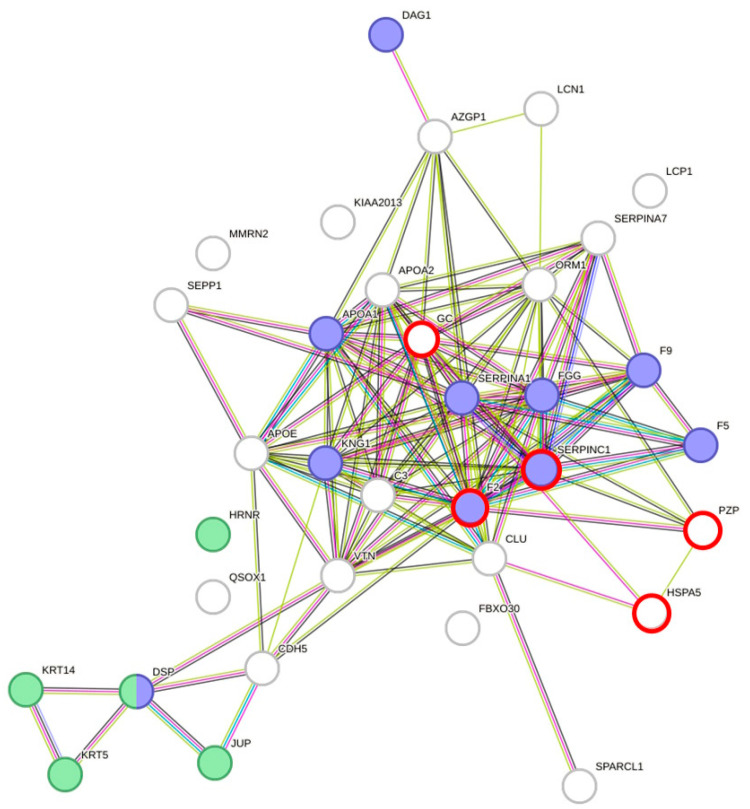
Protein–protein interaction clustering and functional annotation of all differentially abundant proteins found in males and females of each comparison (dehydrated vs. control, dehydrated vs. hyperhydrated, and hyperhydrated vs. control); blue = response to wounding (*p* = 1.74 × 10^−8^) and green = keratinization (*p* = 4.58 × 10^−5^). Created with STRING [19] version 11.5. Red circles mark proteins being differentially abundant in males and females.

**Table 1 nutrients-15-03789-t001:** Average age and number of drugs prescribed in the different groups with standard deviation (SD).

Group	Age ± SD	Drugs ± SD
Normal hydrated men (control)	89.0 ± 3.5	8.4 ± 4.6
Hyperhydrated men	78.2 ± 5.8	8.8 ± 5.6
Dehydrated men	80.6 ± 4.7	13.4 ± 2.0
Normal hydrated female (control)	84.0 ± 3.6	10.4 ± 4.4
Hyperhydrated female	80.6 ± 5.1	5.8 ± 7.4
Dehydrated female	86.2 ± 5.9	10.2 ± 3.3

**Table 2 nutrients-15-03789-t002:** Mean normalized protein intensity of significant proteins in female (=F) and male (=M) for dehydrated (=D), hyperhydrated (=H), and control (=C) patients. Differential abundant proteins (DAP) were identified by *p*-values < 0.05 of *t*-tests between dehydrated vs. control, hyperhydrated vs. control, and dehydrated vs. hyperhydrated; *p* < 0.05 is highlighted in grey and black highlights are proteins found significant in both genders.

Female							Male						
Gene Name	Mean			*p*-Value			Gene Name	Mean			*p*-Value		
	DF	HF	CF	DF vs. CF	HF vs. CF	DF vs. HF		DM	HM	CM	DM vs. CM	HM vs. CM	DM vs. HM
** F2 **	20.27	20.24	20.92	0.010	0.201	0.947	F2	20.94	21.97	21.50	0.171	0.153	0.026
** SERPINC1 **	21.04	21.33	21.89	0.028	0.111	0.168	SERPINC1	21.58	22.54	21.74	0.717	0.069	0.046
** PZP **	18.18	19.59	18.33	0.818	0.093	0.039	PZP	20.30	18.55	19.71	0.231	0.019	0.006
** GC **	20.73	20.91	21.40	0.003	0.080	0.412	GC	21.15	22.18	21.38	0.605	0.116	0.014
** HSPA5 **	15.71	15.52	16.21	0.047	0.019	0.019	HSPA5	16.49	17.04	16.53	0.869	0.092	0.042
**QSOX1**	15.40	14.52	15.80	0.407	0.028	0.018	SERPINA1	22.43	21.27	22.14	0.514	0.024	0.026
**C3**	23.28	23.91	23.83	0.015	0.811	0.134	KRT14	19.57	19.79	19.16	0.069	0.047	0.399
**C4b**	16.16	15.38	16.23	0.732	0.012	0.017	APOA1	22.71	22.42	23.25	0.049	0.435	0.769
**KNG1**	19.08	19.47	19.93	0.032	0.239	0.212	APOE	21.58	21.27	21.98	0.281	0.009	0.378
**IGHG2**	20.35	20.22	17.69	0.049	0.146	0.920	APOA2	20.34	21.08	20.59	0.428	0.148	0.048
**VTN**	19.60	19.92	20.14	0.001	0.411	0.287	FGG	22.31	21.68	22.17	0.641	0.139	0.011
**CLU**	20.03	20.11	20.81	0.005	0.027	0.727	ORM1	19.89	18.43	19.23	0.249	0.053	0.025
**F5**	19.22	19.34	20.30	0.038	0.159	0.825	SERPINA7	18.61	19.49	18.92	0.416	0.137	0.029
**F9**	17.32	16.83	18.38	0.047	0.245	0.673	KRT5	19.81	19.92	19.33	0.045	0.002	0.561
**LCN1**	16.62	16.28	16.58	0.855	0.273	0.040	LCP1	15.15	16.32	15.90	0.038	0.701	0.335
**CDH5**	17.10	16.66	17.55	0.207	0.036	0.144	JUP	16.85	16.64	16.23	0.022	0.061	0.247
**SELENOP**	17.93	17.89	18.60	0.007	0.015	0.861	AZGP1	19.57	20.38	19.72	0.731	0.187	0.046
**Krt86**	16.68	16.49	16.63	0.829	0.483	0.023	DAG1	14.61	14.19	14.58	0.909	0.174	0.043
**MMRN2**	15.81	15.42	16.39	0.104	0.019	0.056	SPARCL1	14.32	15.27	14.81	0.152	0.248	0.035
**DSP**	17.30	17.09	17.23	0.816	0.670	0.016	dnaK	15.83	16.80	16.25	0.385	0.236	0.019
							HRNR	18.01	18.24	17.61	0.054	0.047	0.353
							KIAA2013	15.69	15.81	15.35	0.139	0.044	0.419
							FBXO30	16.57	14.87	15.55	0.011	0.657	0.365

**Table 3 nutrients-15-03789-t003:** Main biological processes from Gene Ontology with a false discovery rate (FDR) < 10^−3^, at least 5 involved genes, and *p* < 0.005 found in the network of significant proteins; relevant subtermini included are marked with an arrow.

# Genes	Description	FDR Value	*p*-Value
**27**	**Localization**	**3.10 × 10^−6^**	**1.21 × 10^−9^**
**27**	**Multicellular organismal process**	**7.80 × 10^−5^**	**2.04 × 10^−7^**
**27** 10 7	**Response to stimulus** → Response to Wounding → Blood Coagulation	**0.0011** 1.12 × 10^−5^ 2.80 × 10^−4^	**6.15 × 10^−6^** 1.74 × 10^−8^ 8.63 × 10^−7^
**22**	**Regulation of biological quality**	**1.91 × 10^−5^**	**3.49 × 10^−8^**
**22**	**Negative regulation of biological process**	**0.0012**	**6.96 × 10^−6^**
**21** 5	**Anatomical structure development** → Keratinization	**0.0037** 0.0048	**3.32 × 10^−5^** 4.58 × 10^−5^
**19**	**Protein metabolic process**	**0.0023**	**1.53 × 10^−5^**
**14**	**Cellular component assembly**	**0.0027**	**1.83 × 10^−5^**
**14**	**Regulation of catalytic activity**	**0.0029**	**2.08 × 10^−5^**
**14**	**Immune system process**	**0.0037**	**3.24 × 10^−5^**
**11**	**Cell activation**	**3.30 × 10^−4^**	**1.26 × 10^−6^**
**11**	**Positive regulation of multicellular organismal process**	**0.0113**	**1.30 × 10^−4^**
**10**	**Regulation of body fluid levels**	**8.22 × 10^−6^**	**1.15 × 10^−8^**
**9**	**Positive regulation of cellular component organization**	**0.0135**	**1.70 × 10^−4^**
**8**	**Cell adhesion**	**0.0126**	**1.50 × 10^−4^**
**5**	**Regeneration**	**0.0014**	**8.31 × 10^−6^**

## Data Availability

The proteomics MS data were deposited to the ProteomeXchange Consortium via the PRIDE [18] partner repository with the dataset identifier PXD043728.

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
