# Peer review of "Hydration Status of Geriatric Patients Is Associated with Changes in Plasma Proteome, Especially in Proteins Involved in Coagulation"

_nutrients, 2023, doi:10.3390/nu15173789_

Round 1

Reviewer 1 Report (Previous Reviewer 1)

The authors addressed all my comments and suggestions

Author Response

Again, we would like to thank the reviewer for the quick and thorough assessment and the constructive advises in order to improve our manuscript.

Reviewer 2 Report (New Reviewer)

Thank you for the opportunity to review, the paper seems interesting, I have the following comments:

L10-11 - Add correspondent author information.

L14-15 - The average age should be added.

L17 and L126 – ‘’ LC-MS’’ - is a form of abbreviation. To expand the impact of an article, always develop abbreviations that appear in the text for the first time. The abstract, main text, tables and graphics should be treated as separate parts.

L50 - Add research hypothesis.

L95 – ‘’ –‘’ Delete.

L97 – ‘’ the clinic.’’ - At which clinic? Did the authors mean department (L58)? Should clarify, unify nomenclature.

L108 – ‘’ (see [16] methods 3.2 step 1-3).’’ - Remove ''see'' is enough information in parentheses.

2.5. Data analysis  - Add the effect size and confidence interval to the results. 

Note to all text-graphics and tables. By the titles, remove the ''-''. Add explanations under graphics and tables all used abberation. In addition, create a list of abbreviations at the end of the text.

L217 – ‘’ (see Fig. 2).’’ - Note to all text. Remove ''see''.

L194 – ‘’ sampl.es’’ - Transcript error.

L246 – ‘’ fema.le’’ - Transcript error.

At the end of the text, add all the missing statements as required by the journal.

‘’ ï‚·  Supplementary Materials:

ï‚·  Author Contributions: For research articles with several authors, a short paragraph specifying their individual contributions must be provided. The following statements should be used "Conceptualization, X.X. and Y.Y.; Methodology, X.X.; Software, X.X.; Validation, X.X., Y.Y. and Z.Z.; Formal Analysis, X.X.; Investigation, X.X.; Resources, X.X.; Data Curation, X.X.; Writing – Original Draft Preparation, X.X.; Writing – Review & Editing, X.X.; Visualization, X.X.; Supervision, X.X.; Project Administration, X.X.; Funding Acquisition, Y.Y.”, please turn to the CRediT taxonomy for the term explanation. For more background on CRediT, see here. "Authorship must include and be limited to those who have contributed substantially to the work. Please read the section concerning the criteria to qualify for authorship carefully".

ï‚·  Funding:

ï‚·  Institutional Review Board Statement:

ï‚·  Informed Consent Statement: ….

ï‚·  Data Availability Statement….

ï‚·  Acknowledgments: In this section you can acknowledge any support given which is not covered by the author contribution or funding sections. This may include administrative and technical support, or donations in kind (e.g., materials used for experiments).

ï‚·  Conflicts of Interest: ….’’

https://www.mdpi.com/journal/nutrients/instructions

Author Response

Thank you for the opportunity to review, the paper seems interesting, I have the following comments:

  • We appreciate the positive feedback and thoroughly addressed the requested points. Changes are included in the manuscript using the “Track Changes” function and pointed out in the response letter.

  • L10-11 - Add correspondent author information.
  • We have added the required information (Line 11).

  • L14-15 - The average age should be added.
  • The average age was added as suggested (Line 16).

  • L17 and L126 – ‘’ LC-MS’’ - is a form of abbreviation. To expand the impact of an article, always develop abbreviations that appear in the text for the first time. The abstract, main text, tables and graphics should be treated as separate parts.
  • We value the remark, that all abbreviations, even commonly applied abbreviations, have to be expanded. Therefore, we revised our manuscript for all abbreviations, especially when used in the tables or graphics sections.

  • L50 - Add research hypothesis.

We have implied our hypothesis by formulating our objective. In line with the reviewer´s recommendation, we have now rephrased this section as follows (line 48ff):
The research hypothesis of this study is
I. to find out whether and, if so, to what extent moderate dehydration or hyperhydration affects the blood proteome,
II. which proteins are differentially abundant and could serve as potential biomarkers, and
III. to investigate with which functions these dysregulations are related.

  • L95 – ‘’ –‘’ Delete. Note to all text-graphics and tables. By the titles, remove the ''-''. Add explanations under graphics and tables all used aberration. In addition, create a list of abbreviations at the end of the text.
  • The additional “-“ was deleted also in every other Table or Graphic which is due to the transformation from the original document to the Nutrient version.

  • L97 – ‘’ the clinic.’’ - At which clinic? Did the authors mean department (L58)? Should clarify, unify nomenclature.
  • We would like to thank the reviewer for this hint, because we as authors know what is meant, of course, but it is more difficult for an external reader. For clarification we have changed it to "Department of Geriatrics".

  • L108 – ‘’ (see [16] methods 3.2 step 1-3).’’ - Remove ''see'' is enough information in parentheses.

L217 – ‘’ (see Fig. 2).’’ - Note to all text. Remove ''see''.

  • Each “see” in the manuscript was deleted as suggested.

  • 5. Data analysis - Add the effect size and confidence interval to the results.
  • Since these values should show the significance in addition to the p-value and Table 1 is already very extensive, we have decided to provide an additional Supplementary Table for reasons of clarity, which contains all the information from Table 1 in addition to Effect Size, Standard Deviation and Confidence Intervals. (Line 203-204)

  • L194 – ‘’ sampl.es’’ - Transcript error.

L246 – ‘’ fema.le’’ - Transcript error.

  • Both transcript errors are deleted.

  • At the end of the text, add all the missing statements as required by the journal.
  • Thank you for pointing out the missing sections that need to be added. All missing sections (Author Contributions, Funding, Institutional Review Board Statement, Informed Consent Statement, Data Availability Statement, Conflicts of Interest, Acknowledgments) have been added (line 375 and following).

This manuscript is a resubmission of an earlier submission. The following is a list of the peer review reports and author responses from that submission.

Round 1

Reviewer 1 Report

A crucial point regarding hydration assessment need to be addressed before considering other potential changes. 

1) Previous studies (Dellinger et al. 2021; Silva et al. 2019; Stratton et al. 2021) showed that there is a lack of agreement between different BIA technologies. Particularly, standing BIA analyzers (like the one used in this study) provide different values of resistance, reactance and phase angle in comparison with foot-to-hand technologies, like the one used by Piccoli et al. for developing the reference tolerance ellipses for the general population. Such as ellipses then are only applicable to data collected with the same technologies. Therefore, your comparisons (individual vectors of the participants on the Piccoli's ellipses) are not properly performed since specific ellipses derived from a standing BIA must be used.

2) Another crucial point concerns the use of BIVA zones for classifying the participants in different hydration status. BIVA is not a method for detecting dehydration or iperhydration (Lukaski 2023). Indeed, a characteristic and innovative aspect of BIVA is that it provides soft tissue classification (under, normal, and over) and ranking (more or less than before intervention), comparing the position of an individual vector to a reference population. Again, the application of BIVA can be compromised if inappropriate reference R–Xc graphs are used, leading to evaluations that are difficult to interpret. 

In this study, participants with long vectors that have been classified by the authors as dehydrated can be simply present a low body mass and then a low quantity of fluids. On the contrary shorter vectors can just represent subjects heavier than the average population and then with higher amounts of fluids. This does not mean that these subjects are dehydrated or iperhydrated.